# Role of Receptor Protein Tyrosine Phosphatase β/ζ in Neuron–Microglia Communication in a Cellular Model of Parkinson’s Disease

**DOI:** 10.3390/ijms22136646

**Published:** 2021-06-22

**Authors:** Marta del Campo, Rosalía Fernández-Calle, Marta Vicente-Rodríguez, Sara Martín Martínez, Esther Gramage, José María Zapico, María Haro, Gonzalo Herradon

**Affiliations:** 1Departamento de Ciencias Farmacéuticas y de la Salud, Facultad de Farmacia, Universidad San Pablo-CEU, CEU Universities, Urbanización Montepríncipe, 28925 Alcorcón, Spain; marta.campomilan@ceu.es (M.d.C.); rosalia.fernandez_calle@med.lu.se (R.F.-C.); marta.vicenterodriguez@ceu.es (M.V.-R.); sara.martin.martinez@outlook.es (S.M.M.); esther.gramagecaro@ceu.es (E.G.); 2Departamento de Química y Bioquímica, Facultad de Farmacia, Universidad San Pablo-CEU, CEU Universities, Urbanización Montepríncipe, 28925 Alcorcón, Spain; josemaria.zapicorodriguez@ceu.es (J.M.Z.); maria.harogarcia@ceu.es (M.H.)

**Keywords:** RPTP β/ζ, MPP+, pleiotrophin, midkine, neuroinflammation, microglia

## Abstract

Pleiotrophin (PTN) is a neurotrophic factor that regulates glial responses in animal models of different types of central nervous system (CNS) injuries. PTN is upregulated in the brain in different pathologies characterized by exacerbated neuroinflammation, including Parkinson’s disease. PTN is an endogenous inhibitor of Receptor Protein Tyrosine Phosphatase (RPTP) β/ζ, which is abundantly expressed in the CNS. Using a specific inhibitor of RPTPβ/ζ (MY10), we aimed to assess whether the PTN/RPTPβ/ζ axis is involved in neuronal and glial injury induced by the toxin MPP+. Treatment with the RPTPβ/ζ inhibitor MY10 alone decreased the viability of both SH-SY5Y neuroblastoma cells and BV2 microglial cultures, suggesting that normal RPTPβ/ζ function is involved in neuronal and microglial viability. We observed that PTN partially decreased the cytotoxicity induced by MPP+ in SH-SY5Y cells underpinning the neuroprotective function of PTN. However, MY10 did not seem to modulate the SH-SY5Y cell loss induced by MPP+. Interestingly, we observed that media from SH-SY5Y cells treated with MPP+ and MY10 decreases microglial viability but may elicit a neuroprotective response of microglia by upregulating *Ptn* expression. The data suggest a neurotrophic role of microglia in response to neuronal injury through upregulation of *Ptn* levels.

## 1. Introduction

Parkinson’s disease (PD) is a progressive neurodegenerative disease characterized by dopaminergic neuronal loss in the substantia nigra and intraneural aggregation of alpha-synuclein in Lewy bodies. The pathophysiology of PD is complex and multifactorial and different pathophysiological mechanisms such as mitochondrial dysfunction, oxidative stress and apoptosis may underlie nigrostriatal dopaminergic neuronal degeneration [1,2]. During the last decade, the neuroimmune component in PD has gained attention from the extensive data collected from cell culture, animal models, and postmortem analyses of human PD brains, establishing a dynamic contribution of chronic, sustained neuroinflammation to neuronal degeneration in the early stages of PD [3].

Neuroinflammation is the initial activation of the innate immune system in the central nervous system (CNS) and is initiated by microglial cells, the resident tissue macrophages in the brain. Microglia can be activated by various stimuli such as neuronal death, mechanical injury, or toxins, and they contribute to tissue homeostasis by promoting the elimination of pathogens, dead cells, or other cellular debris [4]. Nevertheless, when the insult persists or these homeostatic mechanisms are dysregulated, it leads to chronic neuroinflammation and disease progression [5].

Accordingly, microglia activation and increased cytokine levels have been observed in postmortem brains from PD patients, in cerebrospinal fluid (CSF) and peripheral blood mononuclear cells (PBMC) [6,7]. In line with these findings, in vivo PET imaging, showed increased binding of the neuroinflammation ligand 18-kDa translocator protein (TSPO) PK-11195 in the brain of PD patients [8]. However, it is currently debated whether this neuroinflammatory response is an initiator, driver, or consequence of PD.

Growing evidence shows an active cross-talk between neurons and microglia, whereby neurons inform microglia about their status and control microglial activation and motility. Similarly, microglia can also modulate neuronal activity [9]. A loss in this bidirectional communication between neurons and glial cells occurs in the brain of PD patients, promoting the neuroinflammatory processes observed in PD [5]. Understanding neuron-microglia cross-talk will be key for developing effective therapies for PD [10].

Neurotrophic factors are molecules involved in neuronal-microglia cross-talk and have been demonstrated to exert neuroprotective effects in animal models of PD [5]. They are mainly produced by neurons and mediate synaptic plasticity, neuroprotection, neurorestoration, and maintenance of neuronal functions [11]. Glial cells, like microglia and astrocytes, can however also secrete these molecules and activate survival signaling pathways in neurons [12].

Pleiotrophin (PTN) is an important neurotrophic factor involved in CNS repair and survival and differentiation of neurons [13]. PTN expression is upregulated in the brain with different underlying neuropathologies characterized by exacerbated neuroinflammatory processes including PD, but little is known about the cell types that overexpress PTN in these pathological conditions [14]. Importantly, PTN has been shown to exert neuroprotective effects on dopaminergic neurons [15].

PTN has pleiotropic effects through different receptors in different organs and contexts, including syndecan-3, anaplastic lymphoma kinase (ALK), neural cell adhesion molecule (NCAM), and Neuroglycan C, which are expressed in neurons [13,14]. Within the CNS, PTN may regulate neuroinflammatory processes through Receptor Protein Tyrosine Phosphatase (RPTP) β/ζ [13,14], which is abundantly expressed in different cells including neurons and microglia [14,16]. PTN binds RPTPβ/ζ and inactivates its phosphatase activity [17], modulating the phosphorylated state of different neuroinflammatory related proteins such as Anaplastic Lymphoma Kinase (ALK) [18] and Fyn kinase [19]. Of note, Fyn is a major upstream signaling mediator of microglial neuroinflammatory processes in PD [20].

Selective inhibitors of RPTPβ/ζ have recently been developed to study the involvement of the RPTPβ/ζ signaling pathway in the modulation of microglial responses and neuroprotection [21,22]. Recent data strongly suggest an important role of RPTPβ/ζ in the neurotrophic actions of resting and primed microglia, and in the microglia-neuron communication [23]. Therefore, modulation of the PTN/RPTPβ/ζ signaling pathway could be a potential target for treating PD and other related neurodegenerative diseases with associated microglia-mediated proinflammatory processes.

In this work, we aimed to study the consequences of the modulation of the PTN/RPTPβ/ζ signaling pathway and its possible influence in neuron-microglia communication using a well-studied acute MPP+ in vitro model of PD, the endogenous inhibitor of RPTPβ/ζ, PTN, and the selective small-molecule inhibitor of RPTPβ/ζ, MY10.

## 2. Results

### 2.1. PTN but Not MY10 Partially Protects against MPP+ SH-SY5Y Cell Toxicity

First, the effect of increasing concentrations of PTN on MPP+-induced SH-SY5Y cell viability loss was tested in the Methyl Thiazolyl Tetrazolium (MTT) assay 24 h after treatment of neuroblastoma SH-SY5Y cells with PTN and/or MPP+. Significant differences were found between group means as determined by one-way ANOVA (Figure 1, F (7, 274) = 205.3, *p* < 0.0001). PTN treatment alone did not induce significant effects. In line with previous findings, 1 mM MPP+ decreased SH-SY5Y cell viability by 40% (*p* < 0.0001) [24]. Of note, co-incubation with PTN (6 and 12 µM) slightly prevented the cytotoxicity induced by MPP+ by ~5–10% (Figure 1, *p* = 0.0110 and 0.0298 respectively).

To further understand whether the effects of PTN were mediated by RPTPβ/ζ, we next tested the effect of the RPTPβ/ζ inhibitor MY10 on cell viability of MPP+-treated SH-SY5Y cells. Significant differences were found between group means as determined by one-way ANOVA (Figure 2, F (7, 492) = 9.622, *p* < 0.0001). MY10 tended to aggravate MPP+-induced cell viability loss but did not reach statistical significance after correcting for multiple testing (Figure 2). However, we observed that treatment with 1 μM and 10 μM MY10 alone significantly reduced neuronal viability in a dose-dependent manner by 17% and 26% (Figure 2, *p* < 0.0251 and *p* < 0.0001, respectively).

### 2.2. Conditioned Media from MPP+-Injured SH-SY5Y Cells Treated with the Rptpβ/Ζ Inhibitor MY10, Aggravates the MPP+-Induced Loss of Microglial Viability

To test the effect of RPTPβ/ζ inhibition on MPP+-induced microglial injury, BV2 microglial cells were stimulated with MPP+ and/or MY10 for 24 h, and cell viability was measured by MTT assay. We noted a significant effect between group means in the viability of BV2 microglial cells (Figure 3a, F (7, 238) = 87.53, *p* < 0.0001). MPP+ induced a significant decrease of BV2 cell viability (~60–70%, *p* < 0.0001). Co-incubation with MY10 did not cause relevant effects on the MPP+-induced microglia cell viability loss. However, treatment with MY10 alone significantly reduced BV2 cells viability at all concentrations of the inhibitor used (Figure 3a, *p* = 0.0361 (0.1 µM), *p* = 0.0007 (1 µM), *p* < 0.0001 (10 µM)).

To investigate the role of RPTPβ/ζ in the cross-talk between MPP+-injured neuron-like cells and microglia, microglia BV2 cells were exposed to conditioned media from SH-SY5Y cells treated with MPP+ and/or MY10 and cell viability was measured. As shown before, media from SH-SY5Y cells was expected to contain the MPP+ and MY10 added 24 h before to the neuronal cells given their stability [23]. Therefore, we were looking for specific differences between BV2 cells treated with MY10 and MPP+ and BV2 cells treated with the media from MPP+/MY10-treated SH-SY5Y cells. ANOVA indicated that the group means were statistically different (Figure 3b, F (7, 367) = 129.7, *p* < 0.0001). We observed a significant decrease in the viability of BV2 cells incubated with the media from MPP+-injured SH-SY5Y cells (~57%, *p* < 0.0001). This was exacerbated by the treatment with the conditioned media from MPP+-injured SH-SY5Y cells with 10 μM MY10 (Figure 3b, *p* < 0.001), an effect that is not observed in BV2 cells directly treated with MPP+ and MY10 (Figure 3a). Treatment of BV2 cells with the conditioned media from SH-SY5Y cells only treated with MY10 also revealed a significant reduction of the viability of microglial cells at all concentrations of the inhibitor used (Figure 3b, *p* = 0.0045 (0.1 µM), *p* < 0.0001 (1 µM), *p* < 0.0001 (10 µM)).

### 2.3. Upregulated Ptn Expression in Microglia Injured by Media Of MPP+-Treated SH-SY5Y Cells

Since PTN is upregulated in the nigrostriatal pathway of patients with PD [25] and is found to be important for CNS repair and for survival and differentiation of dopaminergic neurons [15], and neuroinflammation [22,23,26], we aimed to measure the expression of *PTN* mRNA in SH-SY5Y cells treated with MPP+/MY10. Real-time quantitative PCR data did not show significant differences in the expression of *PTN* mRNA in SH-SY5Y neuroblastoma cells after MPP+ and MY10 treatments (Figure 4a, F (4, 24) = 0.6312, *p* = 0.6450). MY10 (0.1–10 µM) treatment alone did not induce significant effects on *PTN* levels (data not shown).

To establish whether the conditioned media from MY10-treated and MPP+-injured SH-SY5Y cells affect the expression of *Ptn* on microglial cells, we tested the expression of *Ptn* mRNA in BV2 cells incubated with media from MPP+/MY10-treated SH-SY5Y cells. Interestingly, there was an effect of treatment on the levels of *Ptn* in BV2 cells (Figure 4b, F (4, 19) = 2.797, *p* = 0.05). We found a significant upregulation of *Ptn* mRNA expression in BV2 cells injured with conditioned media from SH-SY5Y cells treated with MPP+ (~10-fold increase, Figure 4b, *p* = 0.0465). The same trend was observed in BV2 cells treated with the conditioned media from SH-SY5Y cells incubated with MPP+ and increasing concentrations of MY10 but statistical significance was not achieved (Figure 4b). MY10 (0.1–10 µM) treatment alone did not induce significant effects on *Ptn* levels (data not shown).

## 3. Discussion

Pleiotrophin is a known modulator of neuroinflammation [22,26] expressed in the substantia nigra of PD patients [25]. Its receptor, RPTPβ/ζ, has been shown to modulate LPS-induced microglial activation and to play an important role in the neurotrophic phenotype of microglia and in microglia-neuron communication [23]. Whether PTN/RPTPβ/ζ axis plays a role in the glial responses associated with PD remains unknown.

Neuroblastoma SH-SY5Y cells injured with MPP+ have been a useful in vitro model for studying neurodegenerative events that may occur in PD [27]. However, SH-SY5Y are neuronal-like cells, and the results obtained in this model need to be confirmed in primary neurons. The aim of the present investigation was to evaluate the possible role of the PTN/RPTPβ/ζ axis on the cytotoxicity induced by MPP+ in SH-SY5Y cells and its possible influence in neuron-microglia communication. We observed that PTN partially prevents MPP+-induced cytotoxicity in SH-SY5Y cells. However, although significant, these data need to be taken with caution since the decrease in MPP+-induced cytotoxicity in SH-SY5Y cells treated with PTN was modest (5–10%). PTN binds to different receptors that mediate actions in different organs and pathological conditions, particularly cancer [14]. In the CNS, RPTPβ/ζ has been proposed to be the main mediator driving PTN actions, including the regulation of neuroinflammatory processes [14,23]. Accordingly, in similar experiments to those presented here, it was previously shown that MY10, the specific inhibitor of RPTPβ/ζ, prevents amphetamine-induced toxicity in PC12 cells in a similar manner to PTN [21]. In contrast, we now observed that MPP+-induced cytotoxicity in SH-SY5Y cells might not be modulated by RPTPβ/ζ inhibition. It has to be noted that in the experiments performed with PTN, MPP+ caused a ~40% decrease in cell viability, whereas, in the experiments performed with MY10, the toxin caused a ~25% decrease in cell viability. Although this variability in the effects of 1 mM MPP+ is considered normal according to results published by many groups [27], we do not know how such variability can affect the performance of PTN and MY10, limiting interpretation of the results. Thus, these findings should still be confirmed by comparing PTN and MY10 with the same exact controls (i.e., cells with the same viability decreased). It is, however, tempting to speculate that the partial protective effects produced by PTN might be mediated by others of its known receptors (Figure 5), which needs to be investigated in studies using RPTPβ/ζ knock-down systems. Additionally, inhibition of RPTPβ/ζ did not exert significant effects on MPP+-induced cytotoxicity in BV2 microglial cells. Interestingly, we observed that MY10 alone decreased the viability of both neuronal-like cells and microglia cells, suggesting that RPTPβ/ζ is involved in neuronal and microglial survival.

Recent studies from our group showed that incubation with the conditioned media from microglial cells (in either resting or LPS-activated state) increased the viability of SH-SY5Y cells. This neurotrophic effect was however disrupted when RPTPβ/ζ was first inhibited in microglial cells, suggesting that RPTPβ/ζ plays a role in the neurotrophic phenotype of microglia and in microglia-neuron communication [23]. Whether this is a bidirectional cross-talk (i.e., not only microglia to neuron but also neuron to microglia) remained unknown. Indeed, microglia can be activated by various stimuli including neuronal injury [4], which is an important mechanism involved in PD pathogenesis [28]. We observed that the conditioned media from MPP+-injured SH-SY5Y cells reduced microglia viability in a similar manner to that found in microglial cells incubated with the toxin. However, this detrimental effect was potentiated when RPTPβ/ζ was first inhibited in SH-SY5Y cells using high concentrations of MY10 (Figure 5). Taking together with previous evidence [23], the data suggest that RPTPβ/ζ participates in the bidirectional cross-talk between neurons and microglia. Considering that the increase of MPP+ cytotoxicity induced by RPTPβ/ζ inhibition was not observed when both treatments were added directly to microglia cells, these data also suggest that high concentrations of MY10 made the MPP+-injured SH-SY5Y cells release additional mediators to the media that are harmful to microglia.

It has been proposed that microglia respond to signals from injured neurons in PD and other diseases [29]. In our studies, we found a significant upregulation of *Ptn* mRNA expression in microglial cells incubated with the conditioned media from MPP+-injured SH-SY5Y cells (Figure 5), which was not observed in the neuronal-like cells. The data suggest for the first time that PTN is involved in the neuroprotective mechanisms triggered by microglia after neuronal injury.

This study has several limitations including the use of undifferentiated SH-SY5Y cells or the high variability on cell viability across MPP+ experiments. However, this proof-of-concept study suggests that PTN exerts a partial neuroprotective effect against MPP+-induced SH-SY5Y cell cytotoxicity and that such effect does not seem to be mediated by RPTPβ/ζ in this in vitro model. Moreover, the data suggest that RPTPβ/ζ is involved in neuron-microglia communication. Inhibition of this receptor with MY10 on MPP+-injured SH-SY5Y cells triggers signals from these neuron-like cells that aggravate the decrease in microglia viability (Figure 5). In response to such injury, microglia induce a neurotrophic response by upregulating *Ptn* levels. We hypothesize that PTN, a rapidly secreted neurotrophic factor [13], would then protect neurons by acting in some of its other receptors (Figure 5), not through RPTPβ/ζ. Overall, these results underpin the important role that neuronal-microglia cross-talk may play in the development of neurodegenerative disorders.

## 4. Materials and Methods

### 4.1. Cell Lines and Treatments

The RPTPβ/ζ inhibitor MY10 was synthesized as previously described [21] and dissolved in a medium containing 0.25% DMSO. PTN was purchased from Sigma Aldrich (Madrid, Spain). MPP+ was purchased from Sigma Aldrich (Madrid, Spain) and dissolved in the medium at a concentration of 1mM.

SH-SY5Y cells were purchased from ATCC (Barcelona, Spain). BV2 murine microglial cells were a generous gift from Professor Antonio Cuadrado (Instituto de Investigaciones Biomédicas “Alberto Sols” (IIBM), Madrid, Spain). Both SH-SY5Y cells and BV2 cells were routinely maintained in RPMI-1640 medium with fetal bovine serum (10%), penicillin (100 U/mL), streptomycin (100 μg/mL) and L-glutamine (4 mM) at 37 °C in 5% CO_2_ humidified air following conditions used by others.

#### 4.1.1. Neuroblastoma SH-SY5Y Cells

Prior to each experiment, undifferentiated SH-SY5Y cells were grown for 24 h on 96-well plates at a concentration of 1 × 10^4^ cells per well for MTT assays or on 24-well plates at a concentration of 1 × 10^5^ cells per well for mRNA extraction and for experiments with conditioned media. SH-SY5Y cells were treated with MPP+ (1 mM) and/or MY10 (or vehicle) at increasing concentrations (0.1 μM, 1 μM and 10 μM) for 24 h. For these cell treatments, MY10 was dissolved in DMSO. Proper controls were added to discard the influence of 0.25% DMSO, the maximum concentration of DMSO used in the final in vitro experiments. After the 24-h incubation period, cell viability was measured by MTT test, or cell lysates were obtained for mRNA extraction.

#### 4.1.2. BV2 Microglial Cell Cultures

To test the possibility that MPP+ and/or MY10 may exert direct effects on microglia, BV2 cells were incubated with both MPP+ and/or MY10. For this purpose, BV2 cells were grown on 96-well plates at a number of 7500 cells/well, and the treatments were performed exactly as in the case of SH-SY5Y cells. After the 24-h incubation period, the viability of BV2 cells were tested by means of the MTT test.

To test the possible communication between injured neuron-like cells and reactive microglia, BV2 cells were grown for 24 h on 96-well plates at a number of 3.5 × 10^3^ cells per well. Then, BV2 cells were treated with the conditioned media from SH-SY5Y cells treated with MPP+ and/or MY10 to assess the effect, if any, of the release of substances from injured SH-SY5Y cells on these microglial cells. After a 24-h incubation of BV2 cells with the conditioned media from SH-SY5Y cells, BV2 cell viability was tested through the MTT test and cell lysates were obtained for mRNA extraction.

### 4.2. Cell Viability–MTT Assay

Cell viability was assessed using the MTT assay [30]. In brief, cells were incubated in a serum-free medium containing MTT solution in the dark for 4 h at 37 °C. The MTT solution was discarded and 100 μL DMSO was added to each well to dissolve the formazan crystals. The value of optical density was measured at a wavelength of 562 nm using a microplate reader (Versa-Max, Molecular Devices, Sunnyvale, CA, USA). Cell survival was calculated as a percentage relative to the control. All determinations were carried out in six replicates, and five to six independent experiments were performed.

### 4.3. RNA Extraction and cDNA Synthesis

Total RNA from SH-SY5Y cells and BV2 cells was isolated with a commercial kit (Rneasy Mini kit, Qiagen, Valencia, CA, USA) after collecting samples in TRIzol reagent (Thermo Fisher Scientific, Madrid, Spain) according to the manufacturer’s instructions. RNA concentration and purity were assessed by OD measurements at 260 nm and 280 nm on a NanoDrop spectrophotometer. The integrity of RNA was confirmed in 1% agarose gels after electrophoresis. cDNA synthesis was performed using 200 ng of RNA and the Primer script RT Kit (Takara, Madrid, Spain) following the manufacturer’s instructions.

### 4.4. Real-Time Quantitative-PCR Analysis

Real-time quantitative PCR (qPCR) analysis was performed using a CFX96 Real Time System (Bio-Rad Laboratories, Hercules, CA, USA) using the SYBR green RT-PCR method [31] and the following oligonucleotide primers, for *PTN* (human): 5′ACAATGCCGAATGCCAGAAG3′ 5′AGGTTTGGGCTTGGTCAGTT3′; *Ptn* (mouse): 5′TTGGGGAGAATGTGACCTAATTAC3′ 5′GGCTTGGAGATGGTGACAGTTTTC3′; for *RPL30* (human): 5′AAGACGAAAAAGTCGCTGGA3′ 5′AAAGCTGGGCAGTTGTTAGC3′; *Rpl13* (mouse): 5′GGTGCCCTACAGTTAGATACCAC3′ 5′TTTGTTTCGCCTCCTTGGGTC3′ for *GAPDH* (human): 5′TGCACCACCAACTGCTTAGC3′ 5′GGCATGGACTGTGGTCATGAG3′; for *Hprt* (mouse): 5′TGCTCGAGATGTCATGAAGG3′ 5′TATGTCCCCCGTTGACTGAT3′.

Reactions were performed using 1μL cDNA diluted 1:8 in RNAfree sterile H20, 4 μL forward/reverse primers (1 μM) and 5 μL 2x SYBR Premix ExTaq II master mix (TB green premix ExTaq, Takara, Madrid, Spain). All reactions were performed in duplicate, in a total reaction volume of 10 μL. The relative gene expression was normalized against *RPL30* and *GAPDH* (SH-SY5Y cells, human) or against *Hprt* and *Rpl13* (BV2 cells, mouse), used as a reference, and shown as fold change versus a control group. All determinations were carried out in duplicates, and six independent experiments were performed.

## 5. Statistics

Data are presented as mean ± standard error of the mean (S.E.M.). Data obtained from viability tests and qPCR analysis were analyzed using one-way ANOVA considering treatment as a variant. Relevant differences were analyzed by post-hoc comparisons with Tukey’s post-hoc tests. *p* < 0.05 was considered statistically significant. All statistical analyses were performed using Graph-Pad Prism version 8 (San Diego, CA, USA).

## Figures and Tables

**Figure 1 ijms-22-06646-f001:**
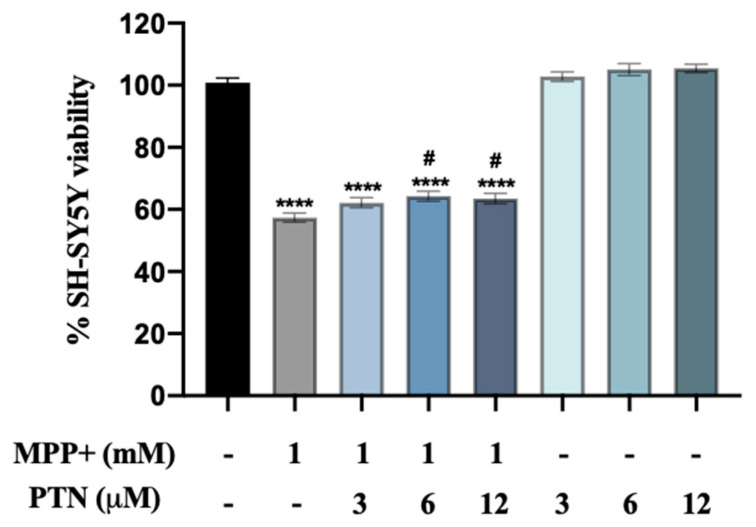
PTN modulates MPP+-induced SH-SY5Y cell toxicity. MPP+ causes a significant decrease in SH-SY5Y viability which is partially prevented by co-incubation with PTN at 6 and 12 µM. **** *p* < 0.0001 vs. Control non-treated cells, # *p* < 0.05 vs. MPP+. Data are shown as % viability considering 100% viability of SH-SY5Y cells incubated with complete media (Control, black bar).

**Figure 2 ijms-22-06646-f002:**
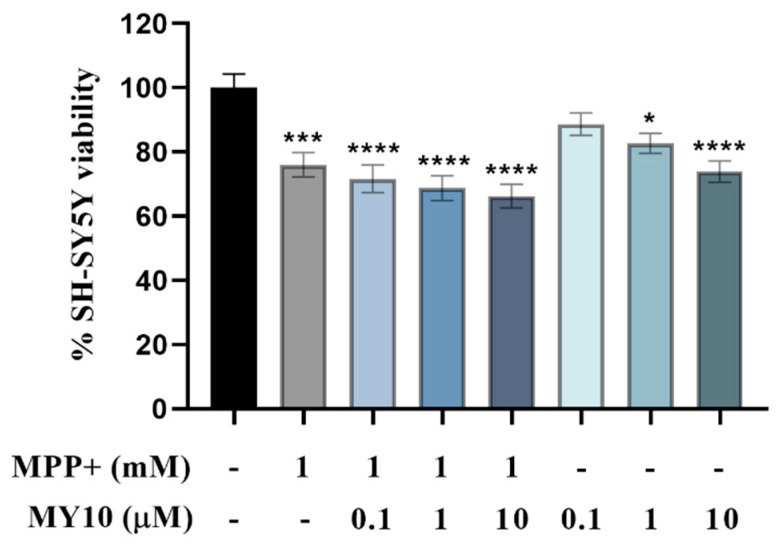
Pharmacological inhibition of RPTPβ/ζ does not modulate MPP+-induced SH-SY5Y cell toxicity but is involved in SH-SY5Y cell viability. MPP+ causes a significant decrease in SH-SY5Y cell viability, but to a lesser extent than that caused by MPP+ in the experiments performed for Figure 1. The toxic effect of MPP+ was not significantly modulated by MY10. However, treatment with MY10 alone caused a significant reduction of SH-SY5Y cell viability at 1 and 10 µM. * *p* < 0.05, *** *p* < 0.001, **** *p* < 0.0001 vs. Control, non-treated cells. Data are shown as % viability considering 100% viability of SH-SY5Y cells incubated with complete media (Control, black bar).

**Figure 3 ijms-22-06646-f003:**
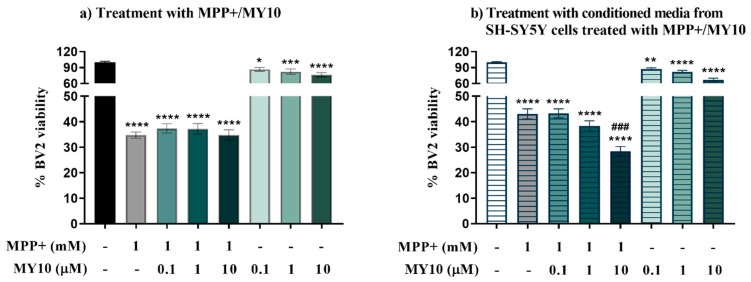
Pharmacological inhibition of RPTPβ/ζ does not modulate MPP+-induced microglia injury but significantly aggravates the loss of microglia viability induced by conditioned media from MPP+-treated SH-SY5Y cells. (**a**) MPP+ causes a significant decrease in BV2 cell viability, which is not significantly modulated by MY10. Treatment with MY10 alone caused a significant reduction of microglial viability. (**b**) MY10 significantly enhances the loss of BV2 cells viability when microglial cells were incubated with the media from MPP+/MY10-treated SH-SY5Y cells. Treatment with conditioned media from SH-SY5Y-treated cells with MY10 alone causes a significant reduction of BV2 cell viability. Data are shown as % viability considering 100% viability of BV2 microglial cells incubated with complete media or with conditioned media from untreated SH-SY5Y cells (Controls, black bar and striped white bar, respectively). * *p* < 0.05, ** *p* < 0.01, *** *p* < 0.001, **** *p* < 0.0001 vs. untreated cells (Control). ### *p* < 0.001 vs. MPP+.

**Figure 4 ijms-22-06646-f004:**
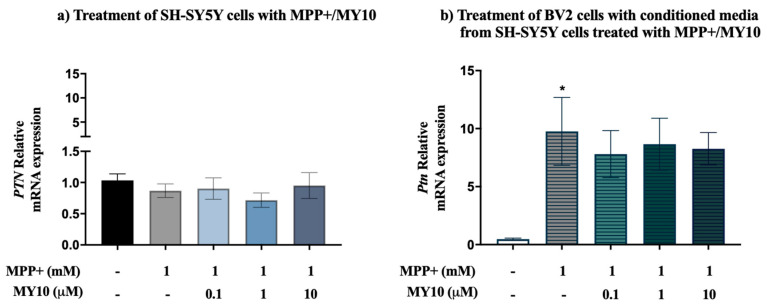
Upregulation of *Ptn* mRNA in microglial cells injured with conditioned media from SH-SY5Y cells treated with MPP+ and MY10. (**a**) Real-time PCR analyses of SH-SY5Y cells incubated with MPP+ and increasing concentrations of MY10 did not reveal significant differences in *PTN* mRNA expression. *GAPDH* and *RPL30* were used as housekeeping genes in human SH-SY5Y cells. (**b**) Real-time PCR analyses of BV2 cells treated with the conditioned media from SH-SY5Y cells incubated with MPP+ and increasing concentrations of MY10 showed *Ptn* mRNA upregulation in BV2 cells. *Rpl13* and *Hprt* were used as housekeeping genes in mouse BV2 cells. * *p* < 0.05 vs. Control.

**Figure 5 ijms-22-06646-f005:**
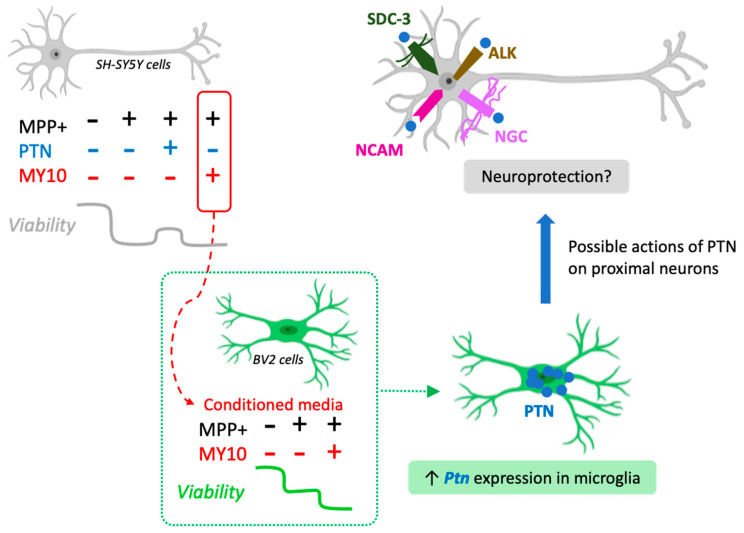
Microglial-induced neurotrophic response. Microglial injury caused by the treatment with conditioned media from SH-SY5Y cells treated with MPP+ is exacerbated when microglial cells are treated with the media from SH-SY5Y cells treated concomitantly with MY10 and MPP+. These treatments significantly induced the upregulation of *Ptn* mRNA in microglial cells. The data suggest a neuroprotective response of microglia through the increased secretion of microglial PTN and enhanced actions of this cytokine in proximal neurons expressing different PTN receptors. This figure was created with BioRender’s web-based software (https://biorender.com/ accessed 14 June 2021). SDC-3: Syndecan-3. ALK: Anaplastic lymphoma kinase. NCAM: Neural cell adhesion molecule. NGC: Neuroglycan C.

## Data Availability

All data available within the manuscript.

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
