# Peer review of "Role of Receptor Protein Tyrosine Phosphatase β/ζ in Neuron–Microglia Communication in a Cellular Model of Parkinson’s Disease"

_ijms, 2021, doi:10.3390/ijms22136646_

Round 1
Reviewer 1 Report
Del Campo and coauthors focused on the role of the RPTP in the regulation of cell viability induced by MPP+ in microglia-neuron comunication. The authors observed a decrease cytotoxicity induced by PTN treatment that can not be associated to the inhibition of RPTP as the MY10 inhibitor of RPTP did not reproduce the same effect. The cytotoxicity of BV2 cells was exacerbated by SH-SY5Y-MPP+/MY10 treated conditioned media suggesting that the PTN effect is not mediated by RPTP and the receptor participate in the crosstalk between microglia and neuron. However, the present study only partially improved from previous evidence and the revision of manuscript is necessary.
- At line 44 it is mentioned that increased levels of cytokines have been observed in PD brain. Please mention also the levels in PBMC and CSF.
- In the figure 1 the data of PTN treatment alone should be added as it has been shown for the correspondence treatment of MY10.
- The housekeeping used in the Figure 4 should be specified in the figure legend.
- It is not clear why the PTN levels have been analysed in BV2 cells treated with conditioned media and not in cells directly treated with MPP+/MY10. Moreover, the protein levels should be analysed as well.
- The authors suggested that the neuroprotective role exerted by microglia after neuronal injury is mediated by PTN. However, only the cell viability and the PTN levels have been described. The microglia-neuron crosstalk is important for neurodegenerative disorders, as described by other more relevant paper in literature. For this reason the conclusion is not completely original and the novelty should be better specified.
- The number of replicates and independent experiments should be added also for the real time experiments.
- The format of PTN gene should be consistent throughout the manuscript.
- The number of cells in chapter 4.1.1 and 4.1.2 must be correct with the exponent.
- SH-SY5Y are neuronal-like cell types. In the text they have been improperly called neurons.
Reviewer 2 Report
Campo et al. have shown that Ptn and MY10, both of which inhibit the RPTP β/ζ, have contrast effects on the survival of SHSY5Y and BV2 cells. This study looks into the role of Ptn-RPTP β/ζ signaling in the neuron-microglia communication in inflammation associated with neurodegenerative diseases, such as Parkinson's Disease.
Major:
1) Ptn is an endogenous inhibitor to the RPTP β/ζ receptor, while MY10 is a synthetic inhibitor of the same. The authors observe that they have contradicting effects on SHSY5Y cells. However, there is no further investigation on this. This could be due to different dissociation coefficients of the inhibitors to the receptor, or could be due to differential influence on the receptor half-life. The latter could be simply checked by looking at the levels of the RPTP β/ζ in the presence of MPP+ with/without Ptn in comparison to MPP+ with/without MY10 in SHSY5Y and BV2 cells.
- This is critical to be addressed before concluding that RPTP β/ζ signaling is not involved in neuron-glia communication.
2) The decline in SHSY5Y viability with MPP+ treatment as compared to Control in Fig 1 and 2 is uneven. It is 40% decrement in Fig.1, while there is only 20% in Fig.2. Effectively, the highest conc of Ptn brings it to ~50% of the control (Fig.1), which is similar to the the highest conc of MY10 in Fig.2. So, there is a good chance that actually Ptn & MY10 have similar effects on SHSY5Y viability, overall. The authors are requested to address/comment to clarify this please.
3) In the discussion, the authors state " our data demonstrate that PTN exerts a partial neuroprotective effect against MPP+-induced neuronal cytotoxicity and that such effect does not seem to be mediated by RPTPβ/ζ in this in vitro model"
- This is not sufficiently supported by the presented data. This can only be verified through Ptn treatment in combination with MPP+ in a RPTPβ/ζ knock-down or knock-out system.
4) Since the aim of the paper is to understand the roe of PTN-RPTPβ/ζ signaling in neuron-microglia communication, it would be best when performed using neuron-microglia co-cultures (PMID: 29117434, 32393376). The authors are strongly urged to perform atleast one key experiment in this system to provide a complete picture.
Minor:
1) Please include the data of Ptn alone on SHSY5Y viability. This is an important condition and needs to be available to the readers for understanding the pattern.
2) The authors need to clarify whether differentiated SHSY5Y cells were used in the experiments. If yes, please mention it in the text, and also include the differentiation protocol in the methods section.
3) In the discussion, the authors mention " Using a specific inhibitor of RPTPβ/ζ (MY10), we now observed that MPP+-induced cytotoxicity in neuronal cells is not modulated by RPTPβ/ζ inhibition, suggesting that the partial protective effects produced by PTN are not mediated by this receptor"
- However, with the presented data, this is an untested assumption. First, the impact of Ptn & MY10 on RPTPβ/ζ could be very different, and need to be tested (Please see point major-1). The other issue is that although SHSY5Y cells are used as a substitute for neurons, the declared statement needs to be verified in primary neurons.
4) Inclusion of the schematic to explain the results is a great effort. However, it would be very useful if the authors improve a bit more on the schematic representation. At present, it is unclear just by itself.
Round 2
Reviewer 2 Report
Major:
Point 2: The discrepancy between the survival rates of undifferentiated SHSY5Y cells with MPP+ treatment in Figs. 1 & 2 can (and needs) to be addressed here. The MPP+ co-treatment with Ptn or MY10 can be performed all on the same set, where the controls are shared between the two inhibitors. Basically, just by doing (atleast) the critical treatments in Figs 1 & 2 at the same time, this issue can be addressed. The authors are strongly recommended to test this. These results can strongly support their claims of differential effects on the RPTP receptor by the two inhibitors.
Minor:
Point 3: The authors have diligently removed all the references of "neuronal" cells from the results section. This is highly appreciated. The authors are further suggested to also clearly state the use of undifferentiated SHSY5Y cells to study neuron-microglia communication as a limitation of this study.
Round 3
Reviewer 2 Report
Dear Authors,
I highly appreciate your response and actions to the comments. I strongly recommend mentioning this discrepancy between the MPP+ controls between Figs 1 & 2 right below the description of Fig 2, in addition to the Discussion.
Author Response
We thank this reviewer for his/her observation. We have now mentioned this discrepancy between the MPP+ controls between Figs 1 & 2 in the legend of Figure 2.